# Assessing Trunk Cross-Section Geometry and Spinal Postures with Noninvasive 3D Surface Topography: A Study of 108 Healthy Young Adults

**DOI:** 10.3390/s25216626

**Published:** 2025-10-28

**Authors:** Arkadiusz Łukasz Żurawski, David Friebe, Sandra Zaleska, Karolina Wojtas, Małgorzata Gawlik, Jacek Wilczyński

**Affiliations:** 1Faculty of Health Sciences, Collegium Medicum, The Jan Kochanowski University in Kielce, 25-317 Kielce, Polandjwilczynski@onet.pl (J.W.); 2Division of Preventive and Sports Medicine, Institute of Occupational, Social and Environmental Medicine, Goethe University Frankfurt, 60590 Frankfurt, Germany; friebe@med.uni-frankfurt.de; 3DIERS International GmbH, 55252 Wiesbaden, Germany; 4Faculty of Management and Computer Modeling, Kielce University of Technology, 25-314 Kielce, Poland

**Keywords:** surface topography, rasterstereography, trunk cross-sectional geometry, spinal posture, noninvasive assessment, reference values, sex differences, young adults

## Abstract

**Highlights:**

**What are the main findings?**
Established normative values for trunk cross-sectional geometry (sagittal/coronal diameters, areas) in healthy young adults, with clear sex-specific differences.Identified modest but significant correlations between thoracic geometry (TorsoScan) and spinal posture parameters (DIERS Formetric).

**What are the implications of the main findings?**
Provides a reference framework for future clinical studies on scoliosis, hyperkypho-sis, and chest wall deformities using noninvasive 3D surface topography.Supports the integrated application of TorsoScan and DIERS Formetric as comple-mentary tools for radiation-free assessment of trunk morphology and spinal alignment.

**Abstract:**

Three-dimensional surface topography offers a noninvasive alternative to radiographic imaging for evaluating trunk morphology and posture. This study aimed to establish normative values of trunk cross-sectional geometry and to examine their associations with spinal alignment in healthy young adults. A cross-sectional sample of 108 participants (62 women, 46 men; 19–23 years, normal BMI) underwent assessment using two complementary techniques: TorsoScan, which reconstructed 360° trunk cross-sections at thoracic levels T1, T4, T8, and T12, and DIERS Formetric 4D, which quantified spinal posture parameters. For each cross-section, sagittal and coronal diameters and cross-sectional areas were calculated; sex differences were analyzed using Welch’s *t* tests and effect sizes, and associations with posture were examined by Pearson correlations with false discovery rate correction and regression modeling. Trunk geometry followed a regular thoracic profile, with the largest coronal diameter at T4 and the maximal area at T8. Men showed consistently larger diameters and areas than women, with large effect sizes at T4–T12. Four associations remained significant: reduced mid-thoracic breadth and area were linked to greater lumbar lordosis, while upper thoracic depth correlated positively with thoracic kyphosis. Predictive models based on posture explained limited variance (R^2^ ≤ 0.19). These findings provide sex-specific reference values and demonstrate that TorsoScan and DIERS Formetric yield complementary, partly convergent measures that may support radiation-free evaluation of trunk morphology in clinical and rehabilitative settings.

## 1. Introduction

The quantitative assessment of trunk cross-sections—sagittal and coronal diameters and cross-sectional areas at defined thoracic levels—has direct biomechanical and rehabilitative relevance. These indices characterize chest wall geometry, which influences spinal alignment, rib mechanics, ventilatory capacity, and overall postural–respiratory function, as well as load distribution in musculoskeletal tissues [1]. Three-dimensional surface topography, particularly rasterstereography, enables reconstruction of trunk and thoracic geometry with high accuracy and reproducibility. Full 360° reconstructions allow automated determination of cross-sectional areas and volumes, differing from computed tomography (CT) by only a few percent, making these indices valuable for objective evaluation of deformities, asymmetries, and therapy-related changes [1]. Because spinal deformities affect chest shape, trunk volume, and body image, cross-sectional metrics are relevant both for planning treatment for scoliosis or hyperkyphosis and for monitoring therapeutic outcomes in functional and aesthetic dimensions—without exposure to ionizing radiation, which is particularly important in youth [1,2,3].

Various studies have established high levels of reliability, as expressed by high values for intraclass correlation coefficients, as well as low values for minimal detectable change, with high degrees of agreement with 3D imaging, which demonstrates high applicability of rasterstereography as a noninvasive, surface-based method for objective evaluation of trunk morphology as a biomarker for clinical purposes, including in rehabilitation studies [4,5,6]. Replicability of measurements with a high level of safety for frequent assessments promotes development of normative data, which cannot be accomplished via radiographic tools as a result of cumulative exposure to radiation [1,3].

Conventional imaging modalities are challenged for repeated, functional trunk imaging. CT requires the use of ionizing radiation, with a doses as low as 50–60 mGy in children, which increases, albeit slightly, the risk of developing leukemia and brain malignancy [7,8]. MRI, on the other hand, uses no radiation, but its costs are high, it is time-consuming, and access is relatively low, with most studies requiring a supine position, which might not represent weight-bearing distribution as accurately [9]. Standup MRI studies are feasible, but most machines are low- to medium-strength machines, which are prone to lower SNR, increased scan time, reducing their functional role [10,11,12]. On a whole, CT faces concerns related to dose restriction, whereas MRI faces issues related to cost, time, as well as supine scanning—thus emphasizing upon non-radiating, rapid, as well as functional-positioning technologies.

TorsoScan (DICAM 3.25.1) is an application based on DIERS Formetric 4D rasterstereography that integrates eight projections to create 360° trunk reconstructions and automatically calculates diameters, areas, and volumes. Validation against CT and phantom models demonstrated small deviations (≈0.5–5%) and excellent reliability (ICC ≥ 0.90), confirming its utility for objective trunk assessment in the standing position [1]. DIERS Formetric 4D itself is a widely used clinical system that projects a structured light grid onto the back surface to derive spinal and pelvic parameters. Its validity has been repeatedly demonstrated, showing high intra- and interday reliability [13], reproducibility of kyphotic and lordotic angles and other postural indices [14], dynamic validity [15], and well-defined SEM/SDC and reference values for monitoring changes [4].

In recent years, rasterstereography has developed from a more specialized tool for diagnosing the spinal column towards a holistic, whole-body technique that could find application in orthopedics, physical rehabilitation, as well as sports medicine. With its connection with digital anthropometry, rasterstereography has allowed for precise 3D analysis of posture, body asymmetries, as well as trunk forms, which in turn helps for the quantitative monitoring of treatments, ergonomic changes, as well as body movement patterns in a more individual, data-driven, nonstatic, yet surface-topography-based understanding of body posture.

Despite increasing evidence on surface topography, no study has correlated 360° trunk cross-sectional measures (sagittal and coronal diameters and cross-sectional area at thoracic levels) obtained with TorsoScan with clinical posture parameters from the DIERS Formetric in healthy adults. The original description of TorsoScan focused on CT validation and reliability of volumes [1], whereas studies on Formetric confirmed reliability in adults without deformities [4] but without reference to 360° trunk geometry. Comparative research between rasterstereography and radiography has focused mainly on adolescent idiopathic scoliosis and conventional angular or rotational indices [5,16], not cross-sections in healthy populations. A literature search in PubMed (August 2025) revealed no studies correlating these two approaches in healthy subjects; to the best of our knowledge, no such evidence has been reported in other databases either. Apart from advancing methodology, there are various possible clinical uses of cross-sectional trunk geometry. For instance, in pulmonology and rehabilitation, trunk geometry could potentially find a role in understanding respiratory function, as it might aid in planning individualized rehabilitation programs for patients with respiratory insufficiencies or related illnesses. Additionally, in orthopedic care, non-invasive analysis of trunk geometry might find a use in monitoring patients with scoliosis, kyphosis, or clients with trunk deformations. Moreover, it might find a role in aesthetic or reconstructive treatments as well. For such uses, it is important to first create a database for normal values for healthy young adults. Consequently, this study sought to examine the relationships between trunk cross-sectional measurements obtained from TorsoScan, as provided in sagittal, coronal diameters, as well as areas at T1, T4, T8, and T12, and DIERS Formetric-measured spinal posture variables for healthy young adults. Additionally, this manuscript sought to address its general objectives as follows: (1) derive a norms value for trunk cross-sectional measurements, as well as sagittal to coronal ratios, (2) analyze gender dimorphism with respect to its influence upon measures of spinal curvature, as well as asymmetries, (3) correlate trunk measurements with DIERS posture variables, as reported in the literature, and (4) create regression models towards predicting trunk-section measurements via indices of posture.

## 2. Materials and Methods

### 2.1. Study Design and Ethical Approval

This was a cross-sectional observational study conducted in accordance with the Declaration of Helsinki. Ethical approval was granted by the Bioethics Committee of Collegium Medicum, Jan Kochanowski University (approval no. 47/2024; decision dated 20 May 2024). All participants provided written informed consent after receiving information about the study purpose, procedures, and potential risks. Confidentiality and anonymity were ensured throughout; participants could withdraw at any time without consequences.

The study was conducted at a single academic center and included participants aged 19–23 years to ensure skeletal maturity and minimize confounding by developmental or degenerative changes.

All image acquisitions were performed by a single trained examiner to avoid inter-rater variability.

### 2.2. Participants

A total of 108 healthy young adults aged 19–23 years with a normal BMI and no diagnosed disorders of the spine or chest wall were included (62 women, 46 men). Clinical screening was performed by a physiotherapist with 13 years of experience in diagnosing and treating postural disorders.

The sample size was determined based on the available population and previous normative rasterstereographic studies, which typically included 80–120 subjects; thus, the sample of 108 participants was considered adequate for descriptive and correlational analysis.

The inclusion criteria were as follows: aged 19–23 years; self-reported good health; normal BMI; and ability to stand unaided.

The exclusion criteria were as follows: history of spinal deformity or surgery; known chest wall deformity; acute musculoskeletal pain; and conditions potentially affecting posture assessment.

The age range (19–23 years) was chosen to obtain a homogeneous group with completed bone growth, stable trunk proportions, and minimal influence of degenerative changes. The goal was to develop reference norms for the young adult population, which would serve as a reference point for further studies in other age groups.

### 2.3. TorsoScan Acquisition (3D Surface Topography; 360° Cross-Sections)

Trunk reconstructions were obtained with TorsoScan (rasterstereography-based). For each participant, eight projections were recorded in quiet standing at 45° increments (0°, 45°, 90°, 135°, 180°, 225°, 270°, 315°). The software performs automatic initial point–cloud alignment (based on shoulder markers), followed by best-fit registration, and generates axial cross-sections at T1, T4, T8, and T12. At each level, the following outcomes were extracted automatically:sagittal diameter (mm);coronal diameter (mm);cross-sectional area (cm^2^).

Although the system enables reconstruction at every vertebral level, four representative thoracic levels (T1, T4, T8, and T12) were selected to capture the upper, middle- and lower thoracic regions while avoiding excessive redundancy (Figure 1).

Participants held their breath mid-respiratory cycle (after exhalation) during scanning. Head, shoulder, and pelvic position were monitored relative to floor markers and the optical plumb line. If deviations > 2° in shoulder rotation or pelvic tilt were detected, the measurement was repeated. All measurements were taken under the same lighting conditions and at the same time of day.

### 2.4. DIERS Formetric Acquisition (Rasterstereography; Posture Parameters)

Spinal posture was assessed via DIERS Formetric 4D in 4D Average mode, which records a 6 s sequence (12 frames at 2 fps) and computes a time-averaged surface to reduce the microsway. Anatomical landmarks (VP—C7; DL/DR—sacral dimples) were identified automatically from surface curvature. The parameters analyzed included the following:Thoracic kyphosis angle VP–T12 (°);Lumbar lordosis angle T12–DM (°);Maximal lateral deviation VP–DM (mm);Maximal surface rotation (°);Trunk length VP–DM (mm).

Definitions and computational details (global vs. peak measures, e.g., ICT–ITL (max)) followed the manufacturer’s documentation.

### 2.5. Standardization and Quality Control

All acquisitions were performed in a habitual bipedal stance: feet on floor markers, knees were extended (without hyperextension), arms relaxed alongside the trunk, and the head was neutral. The light and background conditions were standardized; jewelry and watches were removed; and long hair was tied. The raster projection quality was checked before recording; frames with artefacts were repeated. TorsoScan and DIERS measurements were obtained under identical environmental conditions.

### 2.6. Outcomes

Primary TorsoScan outcomes: sagittal diameter (mm), coronal diameter (mm), and cross-sectional area (cm^2^) at T1, T4, T8, and T12.

Explanatory variables: selected DIERS posture parameters as listed above.

### 2.7. Technical Details of Measurement Systems

All rasterstereographic acquisitions were performed using the DIERS Formetric 4D system (DIERS International GmbH, Wiesbaden, Germany) operated with DICAM software v3.25.1. The system projects a structured white-light grid (640 × 480 pixels, 50 Hz) onto the dorsal surface, and surface curvature is captured by a digital camera located at a fixed distance of 210 cm. The 4D Average mode records a 6 s sequence (12 frames at 2 fps) and computes a time-averaged 3D model to minimize microsway. Anatomical landmarks (VP, DM, DL/DR) were automatically detected using curvature analysis. Calibration was performed before each session using the manufacturer’s reference grid and distance markers, following the DIERS Formetric III manual.

For 360° trunk reconstructions, the TorsoScan module (Formetric-based application) was used. Eight projections were acquired at 45° intervals (0°, 45°, 90°, 135°, 180°, 225°, 270°, 315°) in quiet standing. The software automatically aligned point clouds based on bilateral shoulder markers and generated a continuous volumetric model of the torso. Cross-sections were computed at T1, T4, T8 and T12, and sagittal/coronal diameters and cross-sectional areas were extracted. The system was operated under identical lighting and background conditions for all subjects.

### 2.8. Statistical Analysis

Analyses were performed on the full sample (N = 108). For TorsoScan variables at each thoracic level, descriptive statistics were computed for the whole group and by sex: mean ± SD, median, and range. Sex differences were tested with Welch’s *t* tests; effect sizes were reported as Cohen’s d with 95% confidence intervals (CIs).

Associations between TorsoScan and DIERS parameters were examined via Pearson’s correlations with 95% CIs. To control for multiplicity, the Benjamini–Hochberg false discovery rate (FDR) procedure was applied separately within each family of tests (sex differences, correlations, comparisons of correlations). Significance was set at q < 0.05.

For the prediction models, DIERS variables showing at least moderate bivariate correlation with a given TorsoScan outcome (|r| ≥ 0.30) were considered candidates; sex was included as a mandatory covariate. For predictor selection, Lasso/Elastic Net regularization was used, followed by ordinary least squares (OLS) estimation. Model performance was evaluated via repeated 10-fold cross-validation (5 repetitions, 50 folds in total). The performance metrics included the cross-validated R^2^, root mean square error (RMSE), and mean absolute error (MAE).

For all primary analyses, *p* < 0.05 indicated statistical significance; after multiple comparisons, q < 0.05 (FDR) was used. Effect sizes with CIs were emphasized over *p* values alone. Analyses were conducted in Python 3.13.5 (pandas, numpy, scipy, statsmodels, scikit-learn).

### 2.9. Sample Size and Power

Among the 156 participants screened, 108 were eligible. A priori sample size calculation was not feasible due to the novel nature of the outcomes. Post hoc power analyses indicated very high power (>0.99) to detect large sex differences (d ≈ 1.0–1.3) and high power (~0.90) for moderate correlations (r ≈ 0.30) but only limited power (~0.60) for small-to-moderate effects (d ≈ 0.4). Thus, the study was sufficiently powered for its primary aims (normative values, sex differences, and moderate correlations) but may have been underpowered to detect smaller effects.

## 3. Results

### 3.1. Normative TorsoScan Values

Across thoracic levels, the sagittal diameter increased from T1 (≈129 mm) to T8–T12 (≈217–215 mm), the coronal diameter peaked at T4 (≈427 mm), and the cross-sectional area was maximal at T8 (≈749 cm^2^), with the widest dispersion at T8 (area 404–1530 cm^2^). Detailed descriptive statistics (mean ± SD, median, range, and 2.5th–97.5th percentiles) are provided in Table 1.

### 3.2. Sex Differences

Men presented larger values than women did for nearly all the TorsoScan metrics. After FDR correction, 11/12 comparisons remained significant. The effects were greatest for the coronal diameter and area at T4/T8/T12 (Cohen’s d ≈ 1.0–1.4), e.g., the T8 area differed by ≈ 202 cm^2^, and the T8 coronal area differed by ≈ 63 mm. The full estimates (mean differences, Cohen’s d with 95% CI, *p* and q) are shown in Table 2 and Figure 2.

### 3.3. Associations Between TorsoScan and DIERS Formetric

Among all tested pairs, four correlations remained significant after FDR (Table 3): negative associations between T4 coronal and lumbar lordosis (T12–DM) (r ≈ −0.32), T8 coronal and lumbar lordosis (r ≈ −0.32), T8 area and lumbar lordosis (r ≈ −0.29), and a positive association between T1 sagittal and thoracic kyphosis (VP–T12) (r ≈ +0.30). Partial correlations (controlling for sex) and sex-stratified comparisons (Fisher r→z) did not remain significant after FDR; complete matrices are provided in Appendix A.

### 3.4. Predictive Models for TorsoScan Dimensions

Cross-validated performance was modest. The best model predicted the T8 coronal, with R^2^ = 0.185, RMSE ≈ 45 mm, and MAE ≈ 36 mm; significant predictors included sex (+ ≈ 50.8 mm), pelvic tilt (symmetry line) (β ≈ −2.31 mm/°), and lumbar lordosis (β ≈ +4.46 mm/°). The models for the T8 area and T12 coronal region had R^2^ ≈ 0.07; all the other models had negligible or negative R^2^ values. The cross-validated metrics and coefficients for the significant predictors are reported in Table 4, with the full model sets and diagnostics in Appendix A.

Predictors were preselected from DIERS variables showing at least moderate correlation (|r| ≥ 0.30) with TorsoScan outcomes in the full sample.

## 4. Discussion

This study established normative reference values for trunk cross-sectional geometry in healthy young adults. A typical thoracic pattern was observed, with the maximal coronal diameter at T4 and the maximal cross-sectional area at T8. Pronounced sex-related differences were found, with men showing larger coronal diameters and areas, particularly between T4 and T12. Moderate but consistent associations were identified between trunk geometry and sagittal spinal alignment, while predictive models based solely on posture parameters demonstrated limited explanatory power. These results provide a methodological and clinical basis for interpreting thoracic morphology in relation to spinal posture using radiation-free surface-based techniques.

The observed morphology aligns with known thoracic anatomy, where rib spacing and chest perimeter typically reach their maxima in the mid-thoracic region. The directions of the significant associations are biomechanically plausible: greater lumbar lordosis was related to reduced mid-thoracic breadth, whereas greater thoracic kyphosis corresponded to increased anteroposterior depth in the upper thorax. These findings are consistent with posture-dependent rib and sternal mechanics and support the geometric characterization of the trunk derived from surface methods.

Our findings extend previous work on surface topography by demonstrating moderate convergence between 360° cross-sectional metrics (TorsoScan) and spinal alignment indices (Formetric) in a normative cohort. TorsoScan has demonstrated small deviations compared with CT and excellent reliability for surface-derived areas and volumes [1]. The DIERS Formetric system provides reproducible sagittal and transverse posture measures with high intra- and interday reliability and acceptable measurement error in both static and, partially, dynamic conditions [4,13,14,15]. Multicenter studies have also documented strong agreement between surface topography and radiography for spinal metrics [5,17]. Although our study did not include a radiographic comparison, these previous validations suggest that combining 3D surface data with radiographic benchmarks could further enhance the clinical interpretability of such noninvasive methods.

The greater diameters and areas observed in males at T4–T12 are consistent with established sexual dimorphism of the thoracic cage and ribs [18]. This reinforces the need for sex-specific normative datasets. Evidence from digital anthropometry confirms that surface-derived linear and areal measurements are valid and interpretable, especially when normalized for body size or composition [19,20,21]. In our cohort, part of the variance in area and diameter was attributable to sex; however, incorporating explicit anthropometric parameters such as height, body mass, or body surface area would improve future analyses and comparability across studies.

The present study focused on a narrow age range (19–23 years) to ensure full skeletal maturity and minimize confounding from developmental or degenerative changes. However, relationships between trunk geometry and posture may differ in adolescents or older adults due to variations in thoracic wall compliance, intervertebral disc properties, or rib motion patterns. Future research should therefore explore age-dependent reference values and evaluate how mechanical competence of the thoracic wall evolves across the lifespan.

Potential measurement errors inherent to surface-based techniques include soft-tissue displacement, respiratory motion, and subtle posture deviations. These factors were minimized by standardized positioning—participants stood quietly with relaxed arms and maintained moderate expiration during scanning—while head, shoulder, and pelvic alignment were visually corrected within ±2°. All participants had a normal BMI, which reduced soft-tissue interference. Nevertheless, future studies should integrate tissue-compensation or error-correction models to account for subcutaneous thickness and soft-tissue dynamics.

Cross-sectional indices of the thorax are directly related to the mechanical competence of the chest wall and its ventilatory reserve. In deformity populations, greater structural deviation is associated with diminished pulmonary function, including reduced FVC [22,23]. Radiation-free, repeatable surface imaging methods such as TorsoScan are particularly valuable for young individuals and patients requiring frequent follow-up. Combining surface-based geometry with spirometric or optoelectronic data may enable more precise quantification of the effects of physiotherapy, orthotic correction, or surgical intervention on thoracic and respiratory mechanics.

Strengths of this study include the use of a relatively large normative sample and standardized acquisition using two complementary systems. Limitations include the cross-sectional design, absence of test–retest reliability analysis (SEM/MDC) for this dataset, lack of anthropometric covariates, and single-center setting. The findings therefore apply primarily to healthy young adults with normal BMI. Future studies should incorporate anthropometric normalization (e.g., allometric scaling to body surface area), perform nested cross-validation with sex-specific models, and extend analyses to clinical populations with scoliosis, kyphosis, or chest wall deformities [24]. Additional work is also warranted to develop percentile charts and minimal detectable change thresholds for cross-sectional metrics.

Finally, geometry-based classification frameworks—similar to posture typologies derived from Formetric [25]—could be expanded to include trunk and respiratory phenotypes relevant to treatment outcomes or functional capacity.

### Clinical Implications

Cross-sectional trunk geometry provides clinically relevant indices that can complement spinal assessments in rehabilitation, orthopedics, and respiratory care. In scoliosis or hyperkyphosis, repeated noninvasive assessments can support radiation-free longitudinal follow-up. For chest wall deformities, quantitative indices may guide pre- and post-treatment evaluation. In respiratory rehabilitation, thoracic geometry can be integrated with spirometry to monitor functional improvement. Establishing normative values in healthy young adults offers a key reference for clinical applications and supports the use of TorsoScan and Formetric as complementary diagnostic and therapeutic tools.

## 5. Conclusions

This study provides normative reference data for trunk cross-sectional geometry in healthy young adults. The coronal diameter peaked at T4 and the cross-sectional area at T8, with men showing consistently larger values than women, particularly between T4 and T12. These findings emphasize the need for sex-specific standards in postural and morphological assessment.

The TorsoScan and DIERS Formetric systems deliver complementary information: TorsoScan quantifies 360° trunk geometry, while Formetric captures spinal alignment and back-surface posture. Their modest but biomechanically consistent correlations confirm the relevance of integrating both approaches for comprehensive, radiation-free assessment.

Clinically, these normative datasets may serve as benchmarks for evaluating therapeutic outcomes in orthopedics, physiotherapy, and respiratory rehabilitation. Future studies should expand analyses to broader age groups and patient populations, establish minimal detectable change values, and verify the sensitivity of cross-sectional indices to intervention-induced changes.

## Figures and Tables

**Figure 1 sensors-25-06626-f001:**
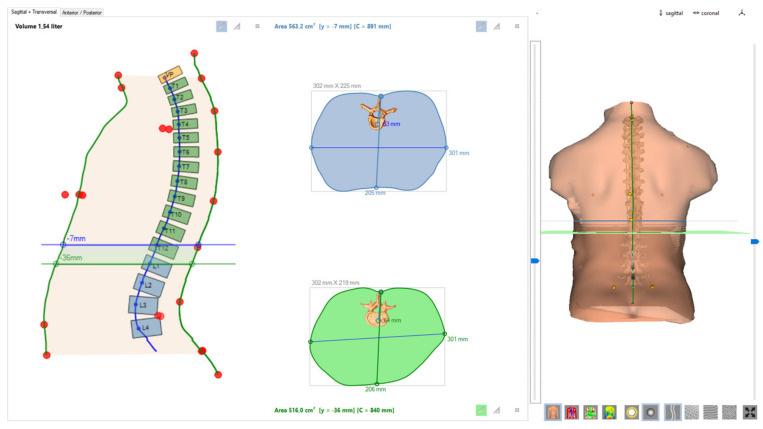
Mage generated using DIERS Formetric 4D/TorsoScan software (DIERS International GmbH, Wiesbaden, Germany).

**Figure 2 sensors-25-06626-f002:**
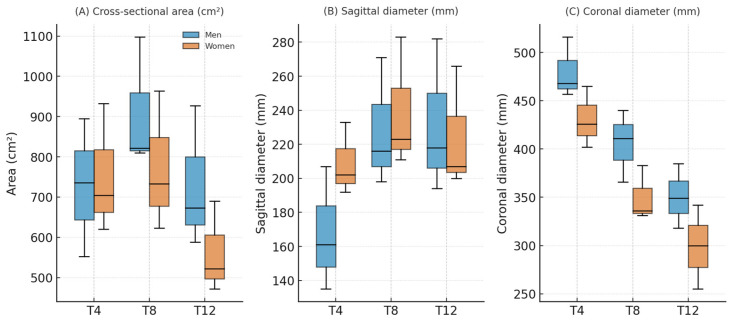
Bloxplots comparing male and female trunk cross-sectional parameters at thoracic levels T4, T8, and T12. (**A**) Cross-sectional area (cm^2^): Men showed consistently larger areas than women at all levels, with the greatest differences at mid- and lower-thoracic levels (T8 and T12). (**B**) Sagittal diameter (mm): The anteroposterior depth of the trunk increased caudally, with men presenting greater values, particularly at T8 and T12. (**C**) Coronal diameter (mm): The transverse width peaked at T4 and declined toward T12; male values exceeded female values at each level. Together, the plots illustrate the typical thoracic morphology—widest at mid-thoracic level—with marked sexual dimorphism across diameters and areas (FDR-adjusted q < 0.05 for 11 of 12 comparisons).

**Table 1 sensors-25-06626-t001:** Normative values of TorsoScan parameters in the entire group (N = 108).

Parameters	N	Mean ± SD	Median	Range (Min–Max)
T1 Sagittal	108	129.34 ± 22.47	128.00	89.00–194.00
T1 Coronal	108	286.06 ± 60.11	280.00	173.00–418.00
T1 Area	108	268.31 ± 101.65	247.45	123.70–613.00
T4 Sagittal	108	169.30 ± 25.77	167.00	122.00–238.00
T4 Coronal	108	426.81 ± 48.87	428.50	162.00–537.00
T4 Area	108	615.62 ± 151.32	594.60	305.20–1130.50
T8 Sagittal	108	216.78 ± 32.64	212.50	157.00–336.00
T8 Coronal	108	372.92 ± 54.99	363.50	258.00–532.00
T8 Area	108	748.58 ± 188.50	720.15	404.10–1530.30
T12 Sagittal	108	215.14 ± 38.92	208.50	107.00–370.00
T12 Coronal	108	310.74 ± 45.34	308.50	223.00–459.00
T12 Area	108	580.95 ± 172.10	545.50	274.60–1403.10

**Table 2 sensors-25-06626-t002:** Sex differences in TorsoScan parameters (men vs women, N = 108).

Parameters	Men Mean ± SD	Women Mean ± SD	Difference (m–w)	Cohen’s d (95% CI)	*p*	q (FDR)	Effect
T1 Sagittal	136.33 ± 22.62	124.16 ± 21.07	+12.17 mm	0.56 (0.17–0.95)	0.0055	0.007	medium
T1 Coronal	297.09 ± 59.02	277.89 ± 60.07	+19.20 mm	0.32 (−0.06–0.70)	0.1003	0.100	small (ns)
T1 Area	293.01 ± 102.71	249.98 ± 97.68	+43.03 cm^2^	0.43 (0.04–0.82)	0.0304	0.034	small
T4 Sagittal	175.61 ± 27.15	164.61 ± 23.85	+11.00 mm	0.43 (0.04–0.82)	0.0311	0.034	small
T4 Coronal	451.20 ± 53.19	408.71 ± 36.35	+42.49 mm	0.96 (0.56–1.36)	1.3 × 10^−5^	<0.001	large
T4 Area	697.39 ± 148.23	554.94 ± 123.16	+142.45 cm^2^	1.06 (0.65–1.47)	8.8 × 10^−7^	<0.001	large
T8 Sagittal	229.37 ± 35.11	207.44 ± 27.41	+21.93 mm	0.71 (0.32–1.10)	0.0007	0.001	medium
T8 Coronal	409.13 ± 46.45	346.05 ± 44.61	+63.08 mm	1.39 (0.97–1.82)	2.3 × 10^−10^	<0.001	very large
T8 Area	864.28 ± 189.97	662.73 ± 134.61	+201.55 cm^2^	1.26 (0.84–1.68)	3.3 × 10^−8^	<0.001	very large
T12 Sagittal	231.22 ± 41.65	203.21 ± 32.21	+28.01 mm	0.77 (0.38–1.17)	0.00028	0.001	medium
T12 Coronal	327.96 ± 39.39	297.97 ± 45.53	+29.99 mm	0.70 (0.31–1.09)	0.00040	0.001	medium
T12 Area	664.98 ± 189.54	518.61 ± 127.19	+146.38 cm^2^	0.93 (0.53–1.33)	2.2 × 10^−5^	<0.001	large

**Table 3 sensors-25-06626-t003:** Significant correlations between TorsoScan and DIERS Formetric parameters (after FDR correction).

TorsoScan Param.	Formetric Param.	N	r (95% CI)	*p*	q (FDR)
T4 Coronal	lumbar lordosis T12–DM [°]	108	−0.32 (−0.48; −0.14)	0.0008	0.027
T8 Coronal	lumbar lordosis T12–DM [°]	108	−0.32 (−0.48; −0.13)	0.0009	0.027
T8 Area	lumbar lordosis T12–DM [°]	108	−0.29 (−0.45; −0.11)	0.0023	0.034
T1 Sagittal	thoracic kyphosis VP–T12 [°]	108	+0.30 (0.11; 0.46)	0.0019	0.034

**Table 4 sensors-25-06626-t004:** Best regression models for TorsoScan dimensions (after cross-validation).

Outcome (TorsoScan)	N	R^2^ (10 × 5)	RMSE	MAE	Significant Predictors (β, *p* < 0.05)
T8 Coronal	108	0.185	44.9	36.0	Sex (+50.8 mm); Pelvic tilt (−2.3 mm/°); T12–DM lordosis angle (+4.5 mm/°)
T8 Area	108	0.068	154.7	119.7	Sex (+164.4 cm^2^); T8 flexion/extension (+48.2); Flèche cervicale (+7.0 mm)
T12 Coronal	108	0.068	40.1	32.5	Flèche cervicale (+1.0 mm); T11 rotation (+3.7 mm)

## Data Availability

The data presented in this study are available in the Appendix A. Additional data are available on reasonable request from the corresponding author.

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
