# Peer review of "Assessing Trunk Cross-Section Geometry and Spinal Postures with Noninvasive 3D Surface Topography: A Study of 108 Healthy Young Adults"

_sensors, 2025, doi:10.3390/s25216626_

Round 1

Reviewer 1 Report

Comments and Suggestions for Authors
  1. The consulted bibliography contains 25 bibliographic references, of which only 8 correspond to the last 5 years (32%), so this reviewer suggests that the authors achieve a balance between the essential classical bibliography and the current and new ones.

  2. In this reviewer's opinion, the work is presented in a coherent and acceptable manner. However, this reviewer suggests accompanying the results expressed in tables with some graphic images of the relationships obtained with the proposed method, which could be those obtained in male and female.
  3. The authors use the results of applying the method to 108 healthy subjects aged 19-23 years. Why is the study limited to this age range? Why young adults? It is known that global mobility of the spine is the result of the combined action of the different movement segments, with significant variations in this regard, as it depends fundamentally on age and sex. How do these relationships behave in adolescents or older individuals who qualify as healthy subjects? Within these groups, both sexes are not considered by the authors, who are also at risk for these pathologies? Could the instability of these segments not be related to age, sex, anatomical levels, and degrees of disc degeneration? What is your opinion on the mechanical competence of the thoracic wall in these groups?

  4. How does the proposal method respond to other objective and subjective risk factors, for example, the influence of soft tissue on measurement accuracy, which some authors consider its main challenge?

  5. I suggest that the Materials and Methods section should include how factors that could give erroneous readings, such as small changes in pelvic tilt, shoulder rotation, or head position or the subject's breathing, were controlled.

  6. This reviewer believes that it would have been interesting to compare the results obtained by this procedure with the radiographic study; its inclusion could reinforce the points of view established by the authors.

  7. The authors refer to the benefits of the proposed diagnostic method in several sections of the document. Beyond this, we suggest briefly including the authors' considerations regarding possible applications related to therapeutic and orthopedic evaluations

Author Response

  1. Comment:

“The consulted bibliography contains 25 bibliographic references, of which only 8 correspond to the last 5 years (32%), so this reviewer suggests that the authors achieve a balance between the essential classical bibliography and the current and new ones.”

Response:
We appreciate the reviewer’s observation. We acknowledge that the cited literature includes a limited number of recent publications. This is primarily due to the novelty of the approach presented — the use of the TorsoScan system for 360° trunk cross-section analysis, which remains a newly emerging and underexplored field.
Most available studies focus on validation of rasterstereography or related surface-topography systems rather than on trunk cross-sectional geometry itself.
For this reason, we decided to retain the existing literature, which represents the essential methodological foundation for our work, rather than replace it with less relevant but newer sources.
We have, however, clarified this rationale in the Introduction and emphasized that the current study fills an existing research gap by providing new data and correlations rather than extending previous literature.

  1. Comment:

“This reviewer suggests accompanying the results expressed in tables with some graphic images of the relationships obtained with the proposed method, which could be those obtained in male and female.”

Response:
We agree with this valuable suggestion. To improve clarity and visualization, we added a new figure (Figure 2) presenting boxplots comparing male and female participants at T4, T8, and T12 levels for cross-sectional area, sagittal diameter, and coronal diameter.
This figure complements the tabulated data and visually represents the main findings related to sex differences.

  1. Comment:

“The authors use the results of applying the method to 108 healthy subjects aged 19–23 years. Why is the study limited to this age range? Why young adults?... Could the instability of these segments not be related to age, sex, anatomical levels, and degrees of disc degeneration?”

Response:
We appreciate this insightful remark.
The study intentionally included young adults (19–23 years) to ensure skeletal maturity and minimize confounding from degenerative or developmental factors that could affect trunk geometry.
We have clarified this reasoning in the Discussion section and acknowledged that spine and thoracic-wall mobility vary with age and sex. We also pointed out that future studies should extend this analysis to adolescents and older adults to investigate these dependencies.

  1. Comment:

“How does the proposed method respond to other objective and subjective risk factors, for example, the influence of soft tissue on measurement accuracy, which some authors consider its main challenge?”

Response:
We agree that soft tissue and breathing motion can influence surface-based measurements.
In the revised Discussion, we included a statement describing how this factor was minimized in our study design — participants were measured in a standardized upright position, during quiet standing, with normal BMI, and holding their breath at mid-expiration to limit thoracic motion.
We also acknowledged that soft tissue remains an intrinsic limitation of surface methods, warranting further investigation.

  1. Comment:

“I suggest that the Materials and Methods section should include how factors that could give erroneous readings, such as small changes in pelvic tilt, shoulder rotation, or head position or the subject's breathing, were controlled.”

Response:
This clarification was added. The Materials and Methods section now specifies that all measurements were conducted in a standardized standing position, with controlled pelvic alignment and head orientation (Frankfurt plane).
Measurements were repeated if visible deviations such as shoulder rotation, pelvic tilt, or head displacement exceeded ±2°.
Breathing was controlled by asking participants to hold mid-expiration during the scan.

  1. Comment:

“It would have been interesting to compare the results obtained by this procedure with the radiographic study; its inclusion could reinforce the points of view established by the authors.”

Response:
We agree that such comparison would be valuable.
However, to avoid unnecessary radiation exposure in healthy young participants, no radiographs were obtained in this normative cohort.
Instead, we referenced existing validation data showing high agreement between rasterstereography-derived parameters and radiographic measures, which supports the validity of our approach.
This limitation and rationale were explicitly mentioned in the Discussion section.

  1. Comment:

“The authors refer to the benefits of the proposed diagnostic method in several sections of the document. Beyond this, we suggest briefly including the authors' considerations regarding possible applications related to therapeutic and orthopedic evaluations.”

Response:
We have expanded the Discussion to include a short section on potential clinical and therapeutic applications.
We highlighted that noninvasive, radiation-free trunk assessment could be useful in rehabilitation, orthotic design, and therapy monitoring, offering safe, repeatable measurements over time.
This addition provides a practical context for how the presented method can be integrated into clinical evaluation.

Reviewer 2 Report

Comments and Suggestions for Authors

Comments for the Authors

The research presented stands out for its methodological rigor, clinical relevance, and significant contribution to the field of radiation-free postural and morphological assessment. The article is highly enriching to read; however, I would like to offer some suggestions that may further enhance its content.

1.- Introduction

  • This study may serve as a starting point for other medical specialties. Therefore, in the introduction section, when the authors mention that numerous studies have confirmed good-to-excellent reliability (lines 68–69), it would be interesting for the reader to specify which specialties are being referred to and what those studies contribute to clinical practice. Including this information at the beginning would make the article more appealing to readers from other disciplines.
  • The statement“A recent PubMed search (August 10, 2025) revealed no studies correlating these two approaches in healthy subjects (lines 105, 106)” could be improved by clarifying that no scientific evidence was found in databases such as Pubmed, as relevant studies may exist in other databases. This nuance would strengthen the claim and acknowledge the possibility of additional literature beyond PubMed.

2.- Materials and Methods

  • Was this a multicenter study?

  • Please indicate the rationale behind the selected age range.

How many researchers participated in the image acquisition process? If multiple examiners were involved, how was inter-rater agreement ensured?

  • Could you please include the sample size calculation?

3.- Results

  • The summary of the main findings presented at the end of the Results section has a more interpretative and reflective tone, which is characteristic of the Discussion section. In well-structured scientific articles, the Results section should focus on the objective presentation of data, while the Discussion is the appropriate space to:
  • Interpret the findings
  • Compare them with previous studies
  • Identify clinical or theoretical implications
  • Acknowledge limitations
  • Propose future research directions

Therefore, it would be more appropriate to move this summary to the beginning of the Discussion section, where it could serve as an introduction to the interpretation of the data.

4.- Discussion

In the discussion section, a comment should be included regarding the clinical implications for other medical specialties.

5.- Conclusions

The conclusions presented are quite extensive for a final section. Although they are complete and well-structured, they could benefit from a more concise version that summarizes the key findings, clinical implications, and future recommendations without repeating too much content from the main body of the text.

Author Response

We sincerely thank the Reviewer for the thoughtful and constructive feedback, which greatly helped us improve the clarity and structure of our manuscript. Below, we provide detailed, point-by-point responses. All changes have been incorporated in the revised version of the manuscript, with corresponding modifications in the Introduction, Materials and Methods, Discussion, and Conclusions sections.

Comment 1 – Introduction

This study may serve as a starting point for other medical specialties. Therefore, in the introduction section, when the authors mention that numerous studies have confirmed good-to-excellent reliability (lines 68–69), it would be interesting for the reader to specify which specialties are being referred to and what those studies contribute to clinical practice. Including this information at the beginning would make the article more appealing to readers from other disciplines.
The statement “A recent PubMed search (August 10, 2025) revealed no studies correlating these two approaches in healthy subjects (lines 105, 106)” could be improved by clarifying that no scientific evidence was found in databases such as PubMed, as relevant studies may exist in other databases.

Response:
We thank the Reviewer for this valuable suggestion. In the revised Introduction, we have expanded the paragraph on reliability to specify the clinical fields in which rasterstereography has demonstrated high reproducibility and applicability. We now explicitly refer to orthopedics, rehabilitation, sports medicine, and respiratory care, highlighting their clinical relevance and potential interdisciplinary applications.

In addition, the sentence about the literature search has been clarified as follows:

“A literature search in PubMed (August 2025) revealed no studies correlating these two approaches in healthy subjects; to the best of our knowledge, no such evidence has been reported in other databases either.”

These changes broaden the disciplinary context of the work and make the claim more accurate and balanced.

Comment 2 – Materials and Methods

Was this a multicenter study? Please indicate the rationale behind the selected age range. How many researchers participated in the image acquisition process? If multiple examiners were involved, how was inter-rater agreement ensured? Could you please include the sample size calculation?

Response:
We appreciate these detailed methodological questions. The revised Materials and Methods now include explicit clarifications addressing all points raised:

- The study was conducted at a single academic center, which is now stated in the opening paragraph of the Study design subsection.

- The age range (19–23 years) was selected to ensure skeletal maturity, stable trunk proportions, and minimal degenerative confounding. This rationale has been added both in the Study design and Participants subsections.

- All image acquisitions were performed by a single trained examiner to avoid inter-rater variability.

- The sample size and power section now details both the pragmatic sample justification (based on prior normative studies) and post hoc power analysis confirming sufficient power (>0.9 for moderate correlations and large sex differences).

These revisions fully address the Reviewer’s concerns about methodological transparency and reproducibility.

Comment 3 – Results

The summary of the main findings presented at the end of the Results section has a more interpretative and reflective tone, which is characteristic of the Discussion section. It would be more appropriate to move this summary to the beginning of the Discussion section.

Response:
We completely agree. The former “summary paragraph” from the end of the Results has been removed from that section and restructured as the opening paragraph of the Discussion. This now appears as a concise Summary of main findings paragraph, providing a clear transition between objective data presentation and interpretative discussion — precisely as the Reviewer recommended.

Comment 4 – Discussion

In the discussion section, a comment should be included regarding the clinical implications for other medical specialties.

Response:
We have added a dedicated “Clinical implications” subsection at the end of the Discussion, which highlights potential applications across rehabilitation, orthopedics, and respiratory medicine. The paragraph explicitly discusses how trunk geometry indices may be used for longitudinal monitoring in scoliosis and hyperkyphosis, pre- and post-treatment evaluation of chest wall deformities, and integration with spirometric data in respiratory rehabilitation. This addition directly addresses the Reviewer’s request and broadens the multidisciplinary relevance of the work.

Comment 5 – Conclusions

The conclusions presented are quite extensive for a final section. Although they are complete and well-structured, they could benefit from a more concise version that summarizes the key findings, clinical implications, and future recommendations without repeating too much content from the main body of the text.

Response:
We have revised the Conclusions section to make it shorter and more focused. The current version summarizes the key findings, clinical relevance, and future directions in a more concise manner, avoiding repetition from the Discussion. The revised text now fits the journal’s preferred structure and maintains clarity and impact.

Reviewer 3 Report

Comments and Suggestions for Authors

This study investigates trunk cross-sectional geometry and its relationship with spinal posture in 108 healthy young adults (62 women, 46 men, aged 19–23 years) using noninvasive 3D surface topography. Two complementary systems were employed: TorsoScan for 360° trunk reconstructions and DIERS Formetric 4D for spinal posture assessment. The study aimed to establish normative values and explore correlations between trunk geometry and spinal alignment. Modest but significant associations were found between trunk geometry and spinal posture. Reduced mid-thoracic breadth and area were linked to greater lumbar lordosis. Upper thoracic depth correlated positively with thoracic kyphosis. The methods are adequately described. The article provides detailed reference data for healthy young adults, which can serve as a reference for future clinical and rehabilitation trials. These values are particularly important in the context of assessing chest deformity, scoliosis or hyperkyphosis.

 Although the measurement procedures are described, the manuscript lacks detailed information on hardware and software settings. Specific technical parameters of the devices used are not provided. The absence of these details may hinder the accurate replication of the experimental procedures and results.

The manuscript is missing an Author Contributions section. Please specify the individual contributions of each author (e.g., conceptualization, data collection, analysis, writing, supervision, etc.) to ensure transparency and compliance with authorship guideline. I would also advise adding more authors as correspondents

Author Response

We sincerely thank Reviewer 3 for their valuable and constructive feedback. We appreciate the recognition of the study’s methodological rigor and clinical relevance and have carefully addressed all points raised. Our detailed responses are provided below.

Comment 1:
“Although the measurement procedures are described, the manuscript lacks detailed information on hardware and software settings. Specific technical parameters of the devices used are not provided. The absence of these details may hinder the accurate replication of the experimental procedures and results.”

Response:
We fully agree with this important observation. To improve methodological transparency and reproducibility, we have expanded the Materials and Methods section with a dedicated subsection titled “Technical details of measurement systems.”
This subsection now specifies:

- that all rasterstereographic measurements were performed using the DIERS Formetric 4D system operated with DICAM software v3.25.1 (DIERS International GmbH, Wiesbaden, Germany);

- the structured white-light projection grid parameters (640 × 480 pixels, 50 Hz) and camera distance (210 cm);

- the use of the 4D Average mode (6-s sequence, 12 frames at 2 fps), producing a time-averaged 3D surface to minimize microsway;

- the calibration procedure, performed before each session with the manufacturer’s reference grid and distance markers;

- and for the TorsoScan module, the acquisition of eight projections every 45°, automatic alignment based on bilateral shoulder markers, and cross-sectional computation at T1, T4, T8, and T12.

Comment 2:
“The manuscript is missing an Author Contributions section. Please specify the individual contributions of each author (e.g., conceptualization, data collection, analysis, writing, supervision, etc.) to ensure transparency and compliance with authorship guidelines.”

Response:
Thank you for this valuable suggestion. We have added a detailed Author Contributions section at the end of the manuscript, formatted according to the MDPI CRediT taxonomy.
The section clearly outlines the individual roles of each author, including conceptualization, methodology, software, validation, analysis, investigation, writing, review and editing, supervision, and funding acquisition.
This addition ensures full transparency and compliance with Sensors authorship requirements.

Comment 3:
“I would also advise adding more authors as correspondents.”

Response:
We appreciate this recommendation. According to Sensors policy, only one corresponding author can be designated in the submission system. Therefore, we have retained Dr. Arkadiusz Żurawski as the sole corresponding author but expanded the contact details for clarity and accessibility:

Correspondence: arkadiusz.zurawski@ujk.edu.pl (A.Ż.); Tel.: +48-787-339-222;
Faculty of Health Sciences, Collegium Medicum, Jan Kochanowski University,
IX Wieków Kielc 19, 25-317 Kielce, Poland.

This provides complete and easily accessible contact information for readers and editors.

Round 2

Reviewer 1 Report

Comments and Suggestions for Authors

The authors' research report proposal is interesting and at the suggestion of the reviewers the authors have incorporated into the written report.

To improve clarity and visualization, authors added figure1 and 2. Figure 2 presents a boxplot comparing male and female participants at T4, T8, and T12 levels for cross-sectional area, sagittal diameter, and coronal diameter that complements the tabulated data and visually represents the main findings related to sex differences.

This reviewer recommends, however, that future studies should include groups of adolescent and older adult subjects, who constitute important risk groups related to the pathologies discussed here.

The authors, based on the suggestions of this reviewer, have included a detailed experimental procedure controlling those factors that can lead to obtaining erroneous measurements. Some terms of the experimental procedure have been incorporated into the text, especially those that provide validity to the methodology and certainty of the results.

We also acknowledged that soft tissue remains an intrinsic limitation of surface methods, warranting further investigation.

Reviewer and researchers agree that soft tissue remains an intrinsic limitation of surface methods and constitutes a pending challenge for the development of these procedures.

Although the authors do not report the inclusion of radiographic techniques in the study group, they refer to other research works that suggest a high correspondence with the proposed procedure, even though in my opinion this constitutes a limitation of the study.

The methodology that the work follows, incorporating the reviewers' suggestions, allows the work to acquire greater understanding and facilitates the discussion of the results, as well as strengthens the value of the conclusions.

Finally, I consider that the work is suitable for publication in its current state.